**Data Availability Statement:** The datasets used and analyzed during the current study are available on biostudies (DOI: 10.6019/S-BSST1768) or upon request from the corresponding author or through

# Specificity of cranial cutaneous manipulations in modulating autonomic nervous system responses and physiological oscillations: A controlled study

**Micha Keller** [1,☯]*, **Volker Perlitz** [2☯], **Holger Pelz** [3], **Stefan Borik** [4], **Ines Repik** [5], **Armin Geilgens** [5], **Birol Cotuk** [6], **Gero Müller** [7], **Klaus Mathiak** [1,8], **Johannes Mayer** [9]

1 Department of Psychiatry, Psychotherapy and Psychosomatics, Medical School, RWTH Aachen University, Aachen, Germany, 2 Simplana GmbH, Aachen, Germany, 3 Deutsche Gesellschaft für Osteopathische Medizin e.V., Buxtehude, Germany, 4 Faculty of Electrical Engineering and Information Technology, Department of Electromagnetic and Biomedical Engineering, University of Zilina, Zilina, Slovakia, 5 Deutsche Gesellschaft für Osteopathische Medizin e.V., Mannheim, Germany, 6 Faculty of Sport Sciences, Department of Sport Health Sciences, Marmara University, Istanbul, Turkey, 7 Raeren, Belgium, 8 JARA-Brain, Research Center Jülich, Jülich, Germany, 9 Deutsche Gesellschaft für Osteopathische Medizin e.V., Augsburg, Germany

☯ These authors contributed equally to this work.
* micha.keller@rwth-aachen.de

## Abstract

Significant autonomic nervous system responses to a specific osteopathic intervention, the cranial vault hold (CVH), have recently been demonstrated in forehead skin blood volume changes, heart rate, and respiration frequencies. The specificity of the CVH-intervention-related autonomic responses yet requires differentiation. Thus, we compared autonomic responses to CVH with responses to compression of the fourth ventricle (CV4) and to two corresponding SHAM conditions. Analysis of frequencies and amplitudes for changes in skin blood volume and respiration in low (LF; 0.05–0.12 Hz), intermediate (IM; 0.12–0.18 Hz), and high (HF; 0.18–0.4 Hz) frequency bands, and metrics of heartrate variability revealed significant decreases in LF range (from 0.12 to 0.10 Hz), increased LF and decreased IM durations, and increased skin blood volume amplitudes in response to CVH, but no significant skin blood volume responses to any of the control interventions. Ratio changes for respiration and skin blood volume frequencies approximately at 3:1 during CVH, remained unchanged in all other interventions. Heart rate decreased across conditions, indicating an increase in parasympathetic tone. This was also indicated by a significant increase in root mean of squared successive difference following CV4. We incurred that rhythmic response patterns in the LF and IM bands only appeared in CVH. This suggests specific physiological responses to CVH warranting further investigation by studying e.g., responses to CVH in physical or mental health disorders with autonomic involvement.

the Deutsche Gesellschaft für Osteopathische Medizin e.V. (Email: kontakt@dgom.info; https://www.dgom.info/).

**Funding:** HEAD Genuit Stiftung (Aachen, Herzogenrath), Grant/Award Number: HGS03S18072016.

**Competing interests:** The authors have declared that no competing interests exist.

**Abbreviations:** OCMM, Osteopathic cranial manipulative medicine; PRM, primary respiratory mechanism; OCF, osteopathy in the cranial field; CRI, cranial rhythmic impulse; ANS, autonomic nervous system; CVH, cranial vault hold; CV4, compression fourth ventricle; IM, intermediate rhythm; PPG, photoplethysmography; SBV, skin blood volume; mTFD, changes, multiscale time-frequency distribution; HRV, heart rate variability; SNA, sympathetic neuron activity; EO, eyes open; EC, eyes closed; retR, reticular rhythm; (LF) (IM) (HF), low frequency band.

## Introduction

Despite practical success and increasing public recognition, osteopathic cranial manipulative medicine (OCMM) or osteopathy in the cranial field (OCF) is still facing controversy. This controversy might reflect confusion originated in the maze of terms used synonymously for the physiological postulate central to OCMM, the primary respiratory mechanism (PRM). This varies from cranial rhythmic impulse (CRI), denoting the PRM palpated at the skull [1], to "breath of life", or "tide" [2]. Common to all terms is a lack of salient objective physiological underpinnings [3].

In their seminal review, McPartland and Mein concluded that any research in this field needs to focus on the autonomic nervous system (ANS) as the most likely origin of such rhythmic phenomena [1]. Following this call, physiological studies presented evidence for rhythmic activity in skin blood velocity at rates similar to those claimed for the CRI or PRM. Relating these oscillations to historic Traube-Hering waves (THW) still fell short of identifying anatomical-physiological correlates accounting for such rhythmicity [4]. Hence, the call to conduct adequate qualitative research to clarify these issues prevailed [5,6]. Yet, the results by Nelson and colleagues and, in particular, those reported recently by [7] hint at possible neurophysiological pathways underlying OCMM.

Interpretation of these earlier findings appears promising when incurring reports on animal experiments suggesting the origin of these oscillations in the lower brainstem region. In these studies, reduction of the arousal level by different interventions (e.g., by i.v. anesthetics or physically by passive movements of the limbs) controlled the emergence of oscillations at approx. 0.15 Hz in non-specific reticular neurons in the lower brainstem. This reticular rhythm (retR) subsequently entrained crucial peripheral physiological systems, respiration, heartbeat, and blood pressure [8], with the rhythm of respiration following the reticular rhythm and not vice versa. This notion gained support by reports of such frequencies in controlled studies in humans. These studies described parallels to the reticular rhythm in animals in that different forms of relaxation provoked the rhythm not only in the skin, but also in respiration, heart rate, and blood pressure [9]. Furthermore, studies on the effects of music showed seemingly contradictory findings since music perceived as stressful amplified this rhythm in the lateral forehead skin [10]. These results have been linked to OCMM, hypothesizing that both the retR described in animal studies and the rhythmic band of 0.15 Hz observed in humans represent the PRM [11].

Subsequently, neurobiological studies on rhythmic activity in the range of 0.15 Hz between 0.1 Hz and respiratory activity showed these intermediate (IM) rhythms associated with central nervous activation of an interoceptive network [12], which appears impaired during neurological stress caused by migraine [13]. Recent functional magnetic resonance imaging (fMRI) studies have provided evidence of a pacemaker based in the brainstem and further supported splitting of the conventionally employed low frequency (LF)-band into a lower (LFa: 0.06–0.1 Hz) and an upper (LFb: 0.1–0.14 Hz) section responsible, e.g., for the processing of anxiety [14,15].

Evidence from these studies suggests inclusion of an IM frequency band with the hitherto used dichotomous analyses of peripheral time series, which rested on widely disputed low frequencies (LF: 0.04–0.15 Hz) and a generally accepted high frequency (HF: 0.15–0.4 Hz) band. Including the IM band not only overcomes rigid frequency border schemes but also accounts for the formation of lower and upper harmonic waves detectable in both the LF- and HF-band, respectively.

A current communication documented the relevance of this approach for understanding ANS responses to a standard technique in OCMM, the cranial vault hold (CVH [16]). Only

including the IM frequency band with LF and HF bands allowed demonstration that CVH as an independent intervention immediately set off two significant changes in rhythmic ANS activity comprising frequencies and amplitudes: one in the 0.15 Hz (IM) band, and another one in the 0.08 Hz (LF) range at T0. Moreover, there was a significant increase of LF activity to IM band activity at T1. While respiration appeared hardly affected at T0, there was an increase to ratios at approx. 3:1 at T1. These responses have been interpreted by these authors as resulting either from amplified reticular network activity or as being related to increased activity in baroreflex pathways.

These findings for CVH-induced increases in LF- and IM-activity, however, raise the question of the specificity of the CVH intervention, because these previously observed but as of yet unexplained reactions in the scalp blood flow [4,17] could theoretically be triggered by different mechanisms [1,7,18]. The latter authors demonstrated that specific osteopathic manipulative treatment (OMT) elicited distinct and specific reductions in BOLD responses in specific brain areas related to interoception, that is, the bilateral insula, anterior cingulate cortex, left striatum, and right middle frontal gyrus [19].

In the current communication, we therefore probed ANS responses in photoplethysmography (PPG), electrocardiogram (ECG), and respiration activity in healthy individuals before, during, and following the application of four interventions in four independent study groups comprising two specific OCMM interventions and two non-specific SHAM interventions. We applied the cranial vault hold (CVH) established in OCMM to trigger PRM/CRI activity, and compression of the fourth ventricle (CV4) established in OCMM to attenuate such activity. As nonspecific interventions, SHAM1 and SHAM2 were designed to control each of these specific interventions. Based on this approach, we hypothesized that physiological reactions in PPG, ECG, and respiration observed during CVH are intervention-specific ANS enhancement reactions, which cannot be triggered equally by an unspecific intervention (SHAM2). ANS responses to CV4 are rather parasympathetically mediated. Therefore, they should lack patterns typical of the IM band physiology. An unspecific intervention mimicking CV4, SHAM1 should exhibit few, if any, differences from CV4.

## Materials and methods

### Experimental protocol and participants

Ninety-eight healthy adults participated in this study. Following the exclusion of measurements owing to extensive artifacts (>1%), technical or other problems, the sample was comprised of eighty-three participants (33 female, 50 male, aged 40.7 ± 12.3 years; Table 1 for details on this sample). In a between-subjects design, the CVH group included 38 participants (56 total measurements, 19 second-day measurements), CV4 comprised 18 participants (29 total measurements, 11 second-day measurements), the SHAM1 had 15 participants (29 total measurements, 14 second-day measurements), and SHAM2 included 12 participants (20 measurements, 8 second-day measurements). Therefore, analyses were performed using 134 measurements. All participants were non-smokers without osteopathic treatment during the preceding three months. Exclusion criteria were mental health symptomatology assessed using the ICD-10 symptom rating (ISR) questionnaire [20], a history of or acute neurological or cardiovascular disorders, current use of psychoactive medication as well as high-performance sports, and smoking. Furthermore, to ensure spontaneous ANS activity, participants were instructed and formally agreed to abstain from caffeine for 4 h and alcohol consumption for 48 h prior to testing. Recruitment was conducted via personal contact. The experimental sessions were held in private practice rooms of four osteopathic practitioners (examiner A: $N = 85$ measurements, examiner B: $N = 17$ measurements, examiner C: $N = 16$ measurements,

**Table 1. Sample characteristics of four groups (cranial vault hold, CVH; compression of 4th ventricle, CV4; SHAM 1; SHAM 2) and tests of statistical difference (ANOVAs) between groups.**

|  | CVH | CV4 | SHAM1 | SHAM2 | Statistics |
|---|---|---|---|---|---|
| Age (M ± SD) | 43.2 ± 12.2 | 40.3 ± 11.5 | 41.6 ± 11.7 | 36.4 ± 12.9 | $F_{(3, 53.3)} = 1.4$, $p = .26$ |
| Sex (female/male) | 18/20 | 5/13 | 5/10 | 5/7 | $X^2_{(3, N = 83)} = 2.3$, $p = .52$ |
| Baseline HR (bpm) | 63.5 ± 10.0 | 71.9 ± 9.0 | 67.4 ± 8.9 | 66.5 ± 9.3 | $F_{(3, 54.3)} = 4.8$, **$p < .01$** |
| Baseline Resp (Hz) | .208 ± .07 | .207 ± .07 | .219 ± .04 | .217 ± .06 | $F_{(3, 58.3)} = .36$, $p = .78$ |
| No. participants | 38 | 18 | 15 | 12 | Total: N = 83 |
| No. measurements | 56 | 29 | 29 | 20 | Total: N = 134 |
| Examiners | A: N = 32 B: N = 14 C: N = 4 D: N = 6 | A: N = 19 B: N = 2 C: N = 4 D: N = 4 | A: N = 19 B: N = 0 C: N = 6 D: N = 4 | A: N = 16 B: N = 1 C: N = 2 D: N = 1 | Total A: N = 86 Total B: N = 17 Total C: N = 16 Total D: N = 15 |

and examiner D: *N* = 15 measurements). The experiment was conducted in accordance with the Code of Ethics of the World Medical Association (Declaration of Helsinki, 2008). All participants provided written informed consent, and all protocols were approved by the Institutional Review Board of the state of lower saxony (Ethics commission vote 07/2017 from 04/03/2017). Recruitment and measurements started on 30.07.2020 and ended on 13.07.2023.

## Examiners

Four DGOM-certified osteopaths participated as examiners in this study. At the time of the study, Examiner A had been in private practice for 34 years and was also teaching OCF. Examiner A estimated that 60% of his patients would undergo cranial treatment, whereas cranial treatment would be the major treatment regimen for approximately 50% of patients. Examiner B had been in private practice for 25 years and was teaching OCF. Examiner B estimated that approximately 80% of his patients would receive at least some cranial treatment, whereas this would be the major treatment for approximately 60% of the patients. Examiners C and D have been active in shared private practices for more than 20 years. Examiner C estimated that 70% of his patients received cranial treatment, and approximately 50% received cranial treatment as the main treatment. Examiner D estimated that 80% of her patients received cranial treatment, and 60% received cranial treatment as the main treatment. To ensure the consistency and reliability of measurements, all osteopathic physicians underwent rigorous training in the application of the four interventions.

## Procedures

All interventions were preceded and followed by sections with open and closed eyes serving as within-subject control conditions (Fig 1). The examiners and participants were blind to data recording and offline data analyses. The sample was merged with our previous CVH measurements, which have been published in part with respect to osteopathic palpation [21] and physiological rhythms during CVH [16]. The participants were randomly allocated to one of the following four interventions.

**Cranial vault hold (CVH).** Standard or cranial vault hold (CVH) is an osteopathic augmentation technique commonly employed in the supine position to stimulate ANS activity by augmenting the primary respiratory mechanism (PRM) [18]. As a standard osteopathic hands-on technique, CVH touches the lateral cranial regions and cranial bones with the palm

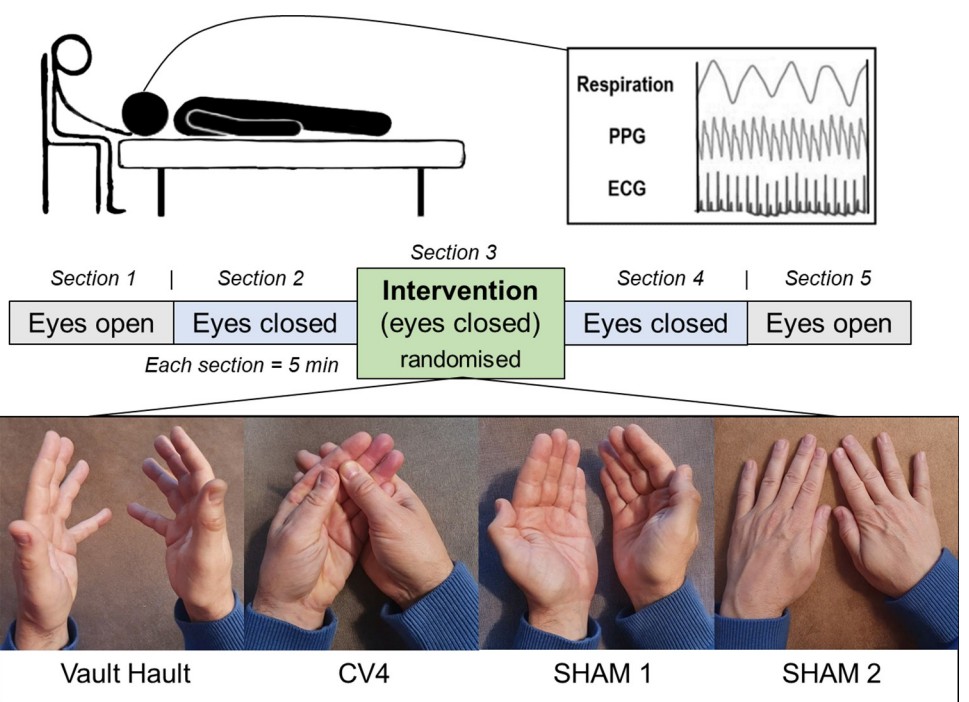

**Fig 1. Experimental paradigm.** Each session was comprised of five sections, lasting 300 seconds each. Section 1 and 5 (eyes open) as well as section 2 and 4 (eyes closed) were performed „hands-off". Section 3 was „hands-on"during which the cranial vault hold was applied and participants' eyes were closed. During each session, respiration, forehead skin blood oscillations (PPG), and electrocardiogram (ECG) were recorded.

of both hands and the palmar surface of the fingers (no thumb) at very low pressure on the scalp (see Fig 1). Notably, in CVH, the index fingers are positioned on the ala majores in order be able to palpate the greatest possible movements.

**Compression of the fourth ventricle (CV4).** Compression of the fourth ventricle (CV4) is an osteopathic suppression technique commonly used in the supine position to suppress the primary respiratory mechanism. Here, the examiners placed one of their hands in the palm of the other so that thenar eminences lay close to one another at an angle of approximately 80˚. With the weight of the head resting on the thenar eminences, the lateral angles of the participants occiput are gently compressed, thereby enhancing cranial extension and attenuating cranial flexion.

**SHAM1.** The CVH technique typically involves bracing the sides of the participant's head. To control for this aspect of the intervention, we implemented the SHAM1 position. This control condition was designed to closely mimic the physical positioning of the CVH technique yet with the instruction to avoid involuntary 'palpating'. The goal was to provide a neutral physical presence without imparting any therapeutic intent or stimulation.

**SHAM2.** During SHAM1, all examiners reported difficulty in maintaining non-palpation, as contact between the palm of their hands and the participant's head was required to mimic the CVH intervention. Consequently, a second sham condition was introduced. To control for potential PRM or ANS stimulation, the participant's head was placed on the back of the examiner's hands. This approach allowed for skin contact between the examiner and participant while avoiding involuntary palpation and stimulation of the participant's head, which could potentially augment PRM or ANS activity.

For the current communication, PPG, ECG (heart rate variability; HRV), and respiration data of participants were used to study physiological rhythms before, during, and following two OCF-specific interventions (CVH and CV4) and the two sham conditions (see Fig 1). During testing, the examiner was seated behind the head of the participant who remained in a supine position. Each experimental session was performed at comfortable room temperature and consisted of five consecutive sections (300 s each; following the first measurements of the study, 30 s were added between sections to account for motion artifacts produced during section transitions). These sections consisted of two different "hands-off" resting states during which the participant was encouraged by an automated female voice to either keep the eyes open (EO) or eyes closed (EC) bracketing the two specific "hands-on" and two SHAM intervention sections during which eyes were kept closed (see Fig 1).

### Physiological data acquisition and preprocessing

**Data acquisition.** During the experimental procedure, physiological signals were recorded using photoplethysmography (PPG), electrocardiography (ECG). Respiratory activity was assessed using accelerometer-based measurements. All recorded data was synchronized using timestamps. The sensors were time-synchronized at the beginning of each measurement with a relative difference of less than 10 ms. The recorded data stream was transmitted to a local computer using Bluetooth, and the complete measurement was uploaded to a secure private cloud for further offline processing.

The PPG device used a standard Osram SFH7060 PG sensor capable of emitting red, infrared, and green light, and which simultaneously detects signals using a built-in photodiode. The data from this sensor was further processed with a microcontroller that captured and digitalized the reflected red-light signals at 660 nm at a sample rate of 125 Hz, while in PPG, we focused only on autonomic nervous system oscillations < 0.4 Hz only. Ambient light was subtracted from the intermediate measurements with all LEDs turned off. ECG data was recorded using a specialized analog digital converter front-end (ADS1292; Texas Instruments, Inc. Dallas, Texas, U.S.) at a sampling frequency of 1000 Hz. Respiration was recorded at a sampling frequency of 500 Hz using a three-axis accelerometer (ADXL354, Analog Devices, Inc., Wilmington, Massachusetts, U.S.), and the resulting respiratory signal was created as the magnitude of the $x$, y-, and z-axis vectors: $r = \sqrt{x^2 + y^2 + z^2}$, where $r$ represents the respiratory signal. For further processing, respiration data was downsampled to 125 Hz.

**Data denoising.** Artifact removal was based on the application of stationary wavelet transform (SWT) [22], in which the raw signals were decomposed into the desired number of levels depending on the type of biosignal. For PPG and respiration data, a 12-level decomposition was used, in which case the SWT could be considered as a band-pass filter. In the case of the PPG, only the approximation component was removed to preserve all signal details; hence, this signal was only detrended with artifact suppression across the entire frequency band based on a moving standard deviation of 1 σ. The sampling frequency was set to 125 Hz. A similar approach was used for the respiration signals. However, only decompositions were selected such that the resulting signal contained only components from 0.03 Hz to 0.5 Hz. For the ECG data, the sampling was left at the original 1000 Hz to preserve the morphological features of the ECG signal, especially the fine-located R-wave position. For the SWT application, only an 8-level decomposition was used, with suppression of the high-frequency components and enhancement of the frequency components related to the QRS complex energy ranging from approximately 2 Hz to 32 Hz.

**Computation of physiological measures.** Before the signal analysis, physiological time series (PPG, ECG, and respiration) were clipped to remove redundant segments. Furthermore,

PPG and respiration data were downsampled to 5 Hz to improve the performance of further data-processing steps.

**Momentary frequency of highest amplitude for PPG and respiration.** Offline processing of the denoised PPG and respiration data was performed using Numpy [23], SciPy [24], and Matplotlib [25]. To further clear the time-frequency plot of the artifacts, the signal was detrended. The preprocessed signal was then converted to a time-frequency distribution using continuous wavelet transformation and a Morlet wavelet (with parameter $\sigma = 5$) computed in relative normalization. The frequency ranged from 0.05 to 0.4 Hz and was sampled in 200 steps. Morlet wavelet time-frequency distributions were computed in relative normalization, which displayed only the frequencies of the highest amplitude of the respective window. 3D-TFD maps were then reduced to 2D-time series of momentary frequencies of the highest amplitude (MFHA), namely the frequency with the highest amplitude for each time step. Frequency band averaged interval durations (total of 300 s for each section) were computed from MFHA values and comprised a single dominant frequency for each time bin. The MFHA was analyzed for the distribution of frequency band activity in previously defined frequency bands (LF: 0.05–0.12 Hz; IM: 0.12–0.18 Hz; HF: 0.18–0.4 Hz; Perlitz et al., 2004).

**PPG signal amplitudes.** A set of algorithms based on the continuous wavelet transformation (CWT) implemented in Matlab R2022a (Mathworks Inc., Natick, MA) was used to extract signal amplitudes for each frequency band (LF: 0.05–0.12 Hz; IM: 0.12–0.18 Hz; HF: 0.18–0.4 Hz). Thus, the amplitudes were extracted for each frequency band from the scalograms computed by continuous wavelet transformation using the time-frequency ridge method. The extracted amplitudes were normalized using min-max scaling for each subject.

**Heart rate variability metrics.** Heart rate variability (HRV) metrics were computed based on the ECG data. All datasets were individually imported into Kubios HRV Scientific 4.0.2 (Kubios Oy, Kuopio, Finland). Automatic noise detection (low), automatic beat correction, and visual inspection were performed before further analysis. We ensured that the automatic beat correction for each experimental section did not exceed 5%. On average, 0.28% of the beats underwent automatic beat correction. The percentage of corrections was not significantly different between sections ($F(4, 249) = 1.0$, $p = .39$). The HRV data were then exported for statistical analysis. Metrics included standard measures, namely the root mean square of successive differences (RMSSD); power in LF, IM, and HF frequency bands; and mean heart rate [26]. The natural logarithm of power in the LF, IM, and HF frequency bands was used because the power values were largely skewed.

## Statistical analyses

For the primary aim, the effects of different interventions (CVH, CV4, SHAM1 and SHAM2) on physiological markers in PPG and respiration of participants were examined via linear mixed model analyses. Several separate models were constructed for different outcome measures (PPG mean frequency of highest amplitude (MFHA), PPG MFHA frequency band durations, PPG amplitudes, HRV metrics, respiratory MFHA). Each model included fixed effects for:

1. **Intervention** (four levels: 1 = CVH, 2 = CV4, 3 = SHAM1, 4 = SHAM2)

2. **Section** (five stages: 1 = eyes open, 2 = eyes closed, 3 = eyes closed + intervention, 4 = eyes closed, and 5 = eyes open)

3. **Time** (two levels: first measurement (T0) and second measurement (T1), separated by at least one week)

4. **Frequency Band** (three levels: low frequency (LF), intermediate frequency band (IM), high frequency band (HF))

Age was included as covariate. To maintain design symmetry and enable post-hoc tests comparing, e.g., the first and fifth experimental blocks, the first block of each measurement (eyes open) was used as outcome variable whenever appropriate. Comparisons with models using the first section as baseline covariate yielded similar results, leading us to choose the previous more comprehensive model. Random intercepts were applied for participants and examiners (1–4) to account for between-subject variability at the starting point. The final model was as follows for PPG and respiration MFHA frequencies as well as all HRV metrics:

$$\text{Dependent variable} \sim 1 + \text{Intervention} + \text{Section} + \text{Time} + \text{Age} + \text{Intervention} : \text{Section} + \text{Intervention} : \text{Time} + \text{Section} : \text{Time} + \text{Intervention} : \text{Section} : \text{Time} + (1|\text{Pat}) + (1|\text{Examiner}).$$

For the model of PPG MFHA durations as well as PPG amplitudes the factor 'band' was added:

$$\text{Dependent variable} \sim 1 + \text{Intervention} + \text{Section} + \text{Band} + \text{Time} + \text{Age} + \text{Intervention} : \text{Section} + \text{Intervention} : \text{Band} + \text{Band} : \text{Section} + \text{Band} : \text{Intervention} : \text{Section} + (1|\text{Examiner}) + (1|\text{Participant}).$$

To further investigate the physiological reactions during intervention, the change in physiology from EC1 (section 2) to intervention was computed, and a linear mixed effects model was used to examine differences between interventions, frequency bands, and time of measurement. For the change analysis, the model specification was:

$$\text{Change dependent variable} \sim 1 + \text{Intervention} + \text{Band} + \text{Age} + \text{Intervention} : \text{Band} + (1|\text{Participant}) + (1|\text{Examiner}).$$

When the normality assumption of residuals was violated, dependent variables underwent appropriate transformations prior to statistical analyses to mitigate heteroscedasticity and reduce skewness. For models with a significant main effect of intervention, section or time, post-hoc pairwise comparison of estimated marginal means was conducted between sections, adjusting for multiple comparisons using the Bonferroni correction. Statistical significance was determined using Kenward-Roger approximation method, which estimates degrees of freedom and calculates $p$-values for mixed-effects models. Results with $p$-values below 0.05 were considered statistically significant.

## Results

### Momentary frequencies of highest amplitude for PPG and respiration

A linear mixed-effects model evaluated the mean MFHA [Hz] of the *PPG* signal across interventions, sections, and time (Fig 2, top row; Table 2, top). No significant main effects were observed for intervention ($F(3, 87.9) = 1.7$, $p = .18$), section ($F(4, 534.5) = 1.0$, $p = .41$), or time ($F(1, 555.6) = .13$, $p = .72$). Age as covariate ($F(1, 60.8) = .78$, $p = .38$) and interactions of section*time ($F(4, 534.7) = .4$, $p = .79$) and intervention*section*time ($F(12, 534.6) = .4$, $p = .95$) were also non-significant. However, a significant intervention*section interaction ($F(12, 534.5) = 3.2$, $p < .001$) was detected. Simple effects analysis revealed significant differences between sections only for the CVH group ($p < .001$). Bonferroni-corrected post-hoc tests for CVH showed significant differences between sections 1–3 ($p = .02$), 2–3 ($p = .03$), 3–4, and 3–5 (both $p < .001$), with lower frequencies observed during section 3. Additionally, a significant intervention*time interaction ($F(3, 516.3) = 4.4$, $p < .01$) suggested differential effects of interventions across timepoints. Simple effects showed a significant difference from the first to

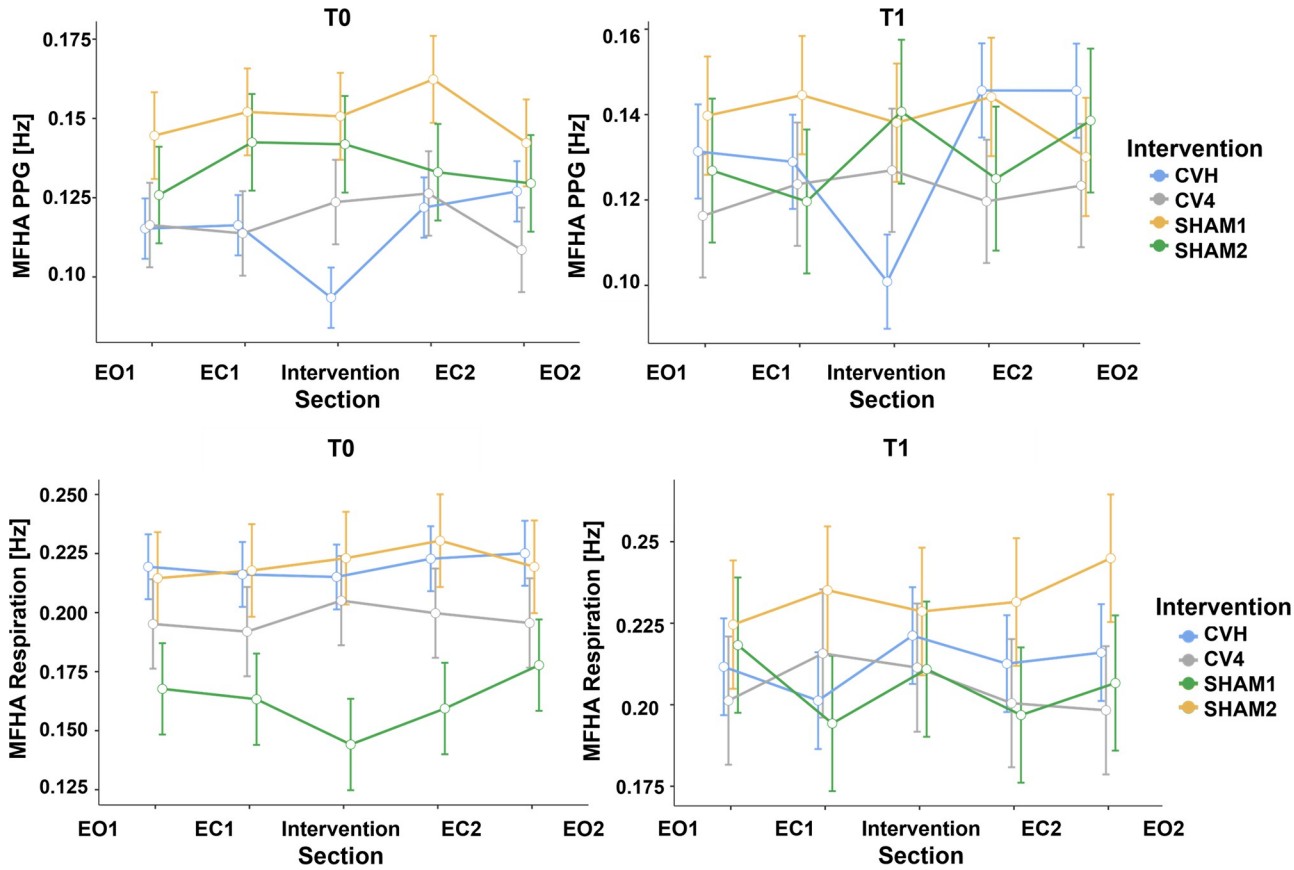

**Fig 2.** Momentary frequency of highest amplitude (MFHA) for photoplethysmography (PPG; top row) and respiratory signals (bottom row) for section (1 to 5), intervention (CVH, CV4, SHAM1, SHAM2) and for first and second (left = T0, right = T1) measurements (Mean ± SE).

second measurement (0.115 Hz to 0.130 Hz) only for CVH ($p < .01$), while other interventions showed no significant changes (all $p > .05$).

To investigate the possibility that *respiratory* activity accounts for changes in PPG activity, a linear mixed-effects model was used to examine the mean respiratory MFHAs (Fig 2, bottom row; Table 2, bottom). This model revealed no significant main effects of intervention ($F(3, 95.2) = 2.1$, $p = .1$) and section ($F(4, 523.7) = .5$, $p = .74$) as well as non-significant interactions for intervention*section ($F(12, 523.7) = .9$, $p = .58$), section*time ($F(4, 524.4) = .7$, $p = .6$) and intervention*section*time ($F(12, 524.4) = .8$, $p = .65$). However, there was a significant effect of time ($F(1, 537.2) = 20.9$, $p < .001$), age ($F(1, 138.6) = 6.6$, $p = .01$) and intervention*time ($F(3, 521.3) = 11.4$, $p < .001$). Simple effects showed a significant increase of respiratory MFHA from the first to second measurement for SHAM1 ($p < .001$) and SHAM2 ($p < .05$), but not for CVH or CV4 ($p > .2$). These results indicate that respiratory frequencies remained stable across sections in each intervention, indicating that observed physiological adaptations in other signals are unlikely attributable to respiratory changes.

## Photoplethysmography MFHA low, intermediate and high frequency band durations

We further computed the *duration of the LF, IM and HF MFHA* (Fig 3A; Table 3). In a linear mixed-effects model, the MFHA durations were investigated with respect to the type of

**Table 2. Momentary frequency of highest amplitude (MFHA) for photoplethysmography (PPG) and respiratory signals in each section (1 to 5), intervention (CVH, CV4, SHAM1, SHAM2) and for first and second (T0, T1) measurements (Estimated marginal mean ± SE).**

| | Section | | Intervention | | | |
|---|---|---|---|---|---|---|
| | | | **CVH** | **CV4** | **SHAM1** | **SHAM2** |
| **PPG MFHA frequency** | **T0** | 1 | 0,115 (±.01) | 0,116 (±.01) | 0,126 (±.02) | 0,145 (±.01) |
| | | 2 | 0,116 (±.01) | 0,114 (±.01) | 0,143 (±.02) | 0,152 (±.01) |
| | | 3 | 0,093 (±.01) | 0,124 (±.01) | 0,142 (±.02) | 0,151 (±.01) |
| | | 4 | 0,122 (±.01) | 0,126 (±.01) | 0,133 (±.02) | 0,162 (±.01) |
| | | 5 | 0,127 (±.01) | 0,109 (±.01) | 0,130 (±.02) | 0,142 (±.01) |
| | **T1** | 1 | 0,131 (±.01) | 0,116 (±.01) | 0,127 (±.02) | 0,140 (±.01) |
| | | 2 | 0,129 (±.01) | 0,124 (±.01) | 0,120 (±.02) | 0,145 (±.01) |
| | | 3 | 0,101 (±.01) | 0,127 (±.01) | 0,141 (±.02) | 0,138 (±.01) |
| | | 4 | 0,146 (±.01) | 0,120 (±.01) | 0,125 (±.02) | 0,144 (±.01) |
| | | 5 | 0,146 (±.01) | 0,123 (±.01) | 0,139 (±.02) | 0,130 (±.01) |
| **Respiration MFHA frequency** | **T0** | 1 | 0,219 (±.01) | 0,195 (±.02) | 0,168 (±.02) | 0,215 (±.02) |
| | | 2 | 0,216 (±.01) | 0,192 (±.02) | 0,163 (±.02) | 0,218 (±.02) |
| | | 3 | 0,215 (±.01) | 0,205 (±.02) | 0,144 (±.02) | 0,223 (±.02) |
| | | 4 | 0,223 (±.01) | 0,200 (±.02) | 0,159 (±.02) | 0,230 (±.02) |
| | | 5 | 0,225 (±.01) | 0,196 (±.02) | 0,178 (±.02) | 0,219 (±.02) |
| | **T1** | 1 | 0,212 (±.01) | 0,201 (±.02) | 0,218 (±.02) | 0,225 (±.02) |
| | | 2 | 0,201 (±.01) | 0,216 (±.02) | 0,194 (±.02) | 0,235 (±.02) |
| | | 3 | 0,221 (±.01) | 0,211 (±.02) | 0,211 (±.02) | 0,229 (±.02) |
| | | 4 | 0,213 (±.01) | 0,200 (±.02) | 0,197 (±.02) | 0,232 (±.02) |
| | | 5 | 0,216 (±.01) | 0,198 (±.02) | 0,207 (±.02) | 0,245 (±.02) |

intervention, section, time and frequency band. In addition to the main effects, the interaction of intervention*section, intervention*band, section*band as well as for intervention*band*section were added to the model.

The main effects for intervention ($F(3, 64.3) = 0.6$, $p = .59$) and time ($F(1, 1250) = 0.0$, $p = .95$), the covariate age ($F(1, 46.4) = .3$, $p = .57$), the interaction of intervention*section ($F(12, 1819.7) = 0.24$, $p = .99$) as well as section*band ($F(8, 1819.7) = 1.59$, $p = .12$) were not significant. However, the main effect of section ($F(4, 1819.7) = 6.2$, $p < .001$), band ($F(2, 1819.7) = 173.1$, $p < .001$) as well as the interactions for intervention*band ($F(6, 1819.7) = 15.4$, $p < .001$) and intervention*section*band ($F(24, 1819.7) = 2.1$, $p = .001$) yielded significant results. Simple effects for sections in separate frequency bands showed a significant effect of section for CVH in the LF and IM ($p < .001$), but not in the HF band ($p = .06$). The simple effects for section were not significant in any other intervention (all $p > .2$). Bonferroni-corrected post-hoc tests for the LF band indicated that the MFHA durations for sections 1, 4, and 5 were significantly lower than those for section 3 ($p = .05$, $p < .001$, and $p < .001$, respectively). Furthermore, there was also a significant decrease in LF MFHA duration from sections 1 to 5 and from sections 2 to 5 (both $p < .001$). In the IM band, section 3 was significantly lower than sections 1, 2, and 4 (all $p < .001$) (Table 3, top).

*MFHA duration change scores* were computed [Section 3 (Intervention)–EC2] to further investigate the effect of the interventions on MFHA duration (Fig 4, left; Table 3, bottom). In this model, the main effect of intervention ($F(3, 64.4) = 0.0$, $p = 1.0$) and covariate age ($F(1, 44.9) = 0.0$, $p = 1.0$) were not significant. This indicates that the change from EC2 to the intervention was not significantly different between the interventions across all frequency bands.

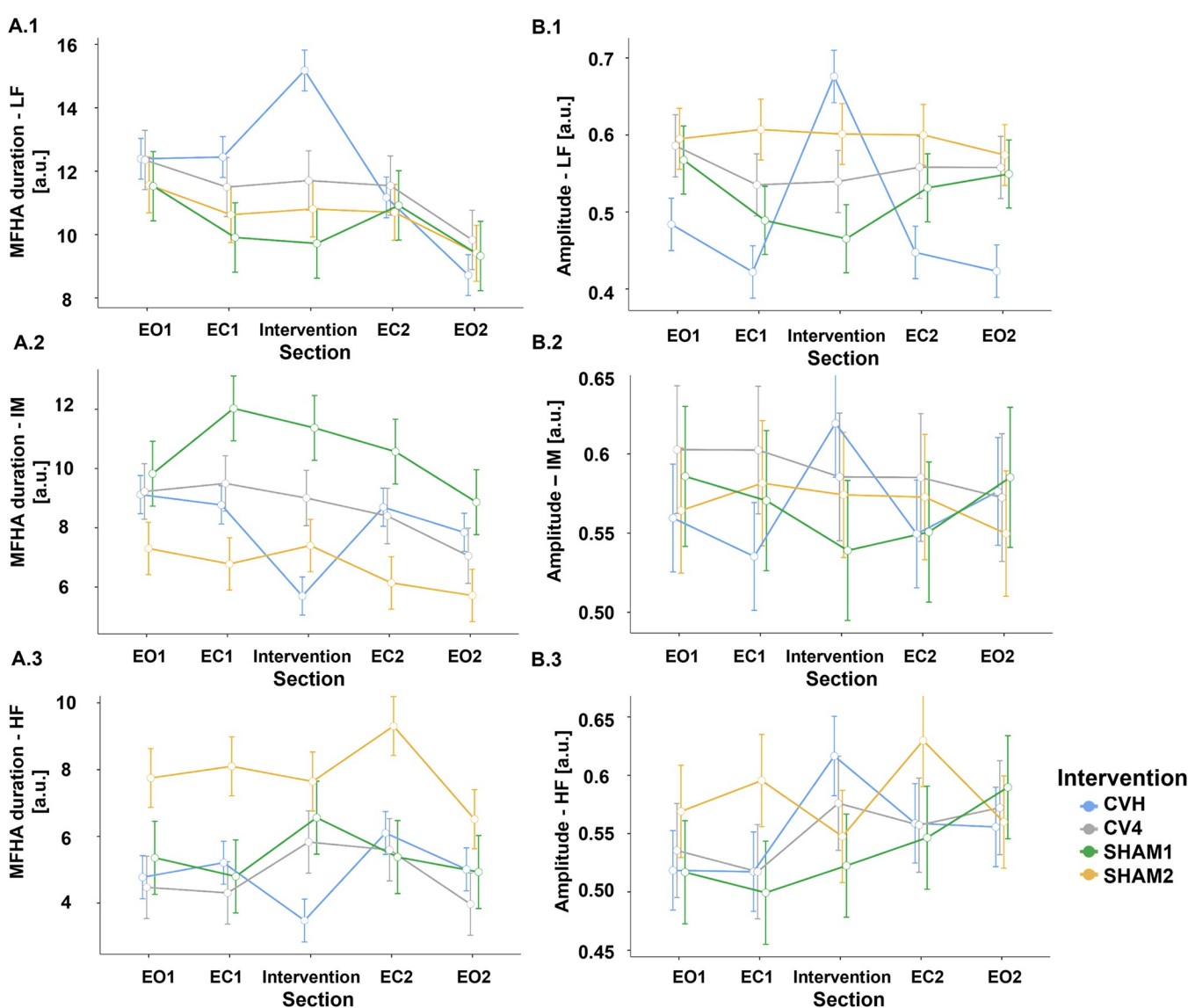

**Fig 3.** Estimated marginal means (± SE) of momentary frequency of highest amplitude (MFHA; A.1-A.3) as well as amplitudes (B.1-B.3) for each intervention, LF, IM, and HF frequency bands, and each section. Due to non-significant main effect of 'time', means across measurement days are plotted. For MFHA durations, CVH exhibits significant concomitant 'shifting' of ANS activity between IM- and LF-band during CVH intervention: LF-band duration increases by the amount that IM-band decreases. During CVH amplitudes in all frequency bands increase.

However, there was a significant effect of band ($F(2, 308.6) = 6.9$, $p = .001$) and a significant intervention*band interaction ($F(6, 308.6) = 9.9$, $p < .001$). A simple effects analysis showed a significant difference between interventions when examining frequency bands separately (LF: $F(3, 315) = 11.7$, $p < .001$; IM: $F(3, 315) = 4.7$, $p < .01$; HF: $F(3, 315) = 2.6$, $p = .05$). Bonferroni-corrected post-hoc tests showed that change scores were significantly higher in LF MFHA durations for CVH than for CV4, SHAM1 and SHAM2 (all $p < .001$), but not between any other interventions (all $p > .2$). For IM MFHA duration change, there was a significant difference between CVH and CV4 ($p = .02$) as well as between CVH and SHAM1 ($p < .001$), while all other differences did not survive correction. Furthermore, the change in HF MFHA duration was significantly different between CVH and CV4 ($p = .02$).

**Table 3. Estimated marginal means (± SE) of durations of momentary frequency of highest amplitude (MFHA).** Top: Mean MFHA duration values for each intervention (CVH, CV4, SHAM1, SHAM2), frequency band (LF, IM, HF), and section (1–5) after transformation [a.u.]. Bottom: Mean change in MFHA duration (±SD) for each intervention and frequency band [in s]. The most marked change can be seen during CVH intervention, with a significant increase in LF and a significant decrease in IM duration.

| | | Section | Intervention | | | |
|---|---|---|---|---|---|---|
| | | | CVH | CV4 | SHAM1 | SHAM2 |
| Frequency band | LF | 1 | 12.4 (±.7) | 12.4 (±.9) | 11.6 (±.9) | 11.5 (±1.1) |
| | | 2 | 12.4 (±.7) | 11.5 (±.9) | 10.6 (±.9) | 9.9 (±1.1) |
| | | 3 | 15.1 (±.7) | 11.7 (±.9) | 10.8 (±.9) | 9.7 (±1.1) |
| | | 4 | 11.1 (±.7) | 11.5 (±.9) | 10.7 (±.9) | 10.9 (±1.1) |
| | | 5 | 8.7 (±.7) | 9.8 (±.9) | 9.4 (±.9) | 9.3 (±1.1) |
| | IM | 1 | 9.1 (±.7) | 9.3 (±.9) | 7.3 (±.9) | 9.9 (±1.1) |
| | | 2 | 8.8 (±.7) | 9.5 (±.9) | 6.8 (±.9) | 12.1 (±1.1) |
| | | 3 | 5.7 (±.7) | 9.0 (±.9) | 7.4 (±.9) | 11.4 (±1.1) |
| | | 4 | 8.7 (±.7) | 8.4 (±.9) | 6.1 (±.9) | 10.6 (±1.1) |
| | | 5 | 7.9 (±.7) | 7.1 (±.9) | 5.7 (±.9) | 8.9 (±1.1) |
| | HF | 1 | 4.8 (±.7) | 4.5 (±.9) | 7.8 (±.9) | 5.4 (±1.1) |
| | | 2 | 5.2 (±.7) | 4.3 (±.9) | 8.1 (±.9) | 4.8 (±1.1) |
| | | 3 | 3.4 (±.7) | 5.8 (±.9) | 7.6 (±.9) | 6.6 (±1.1) |
| | | 4 | 6.1 (±.7) | 5.6 (±.9) | 9.3 (±.9) | 5.4 (±1.1) |
| | | 5 | 5.0 (±.7) | 4.0 (±.9) | 6.5 (±.9) | 5.0 (±1.1) |
| | Mean change in MFHA duration | | Frequency band | | | |
| | | | LF | IM | HF | |
| | | Intervention CVH | 69.5 (±82.3) | -47.0 (±63.3) | -22.4 (±57.7) | |
| | | CV4 | -3.6 (±70.2) | -10.0 (±62.3) | 13.6 (±58.5) | |
| | | SHAM1 | -1.8 (±52.3) | -15.6 (±56.8) | 17.4 (±60.9) | |
| | | SHAM2 | 1.2 (±79.7) | 6.9 (±59.5) | -7.8 (±64.0) | |

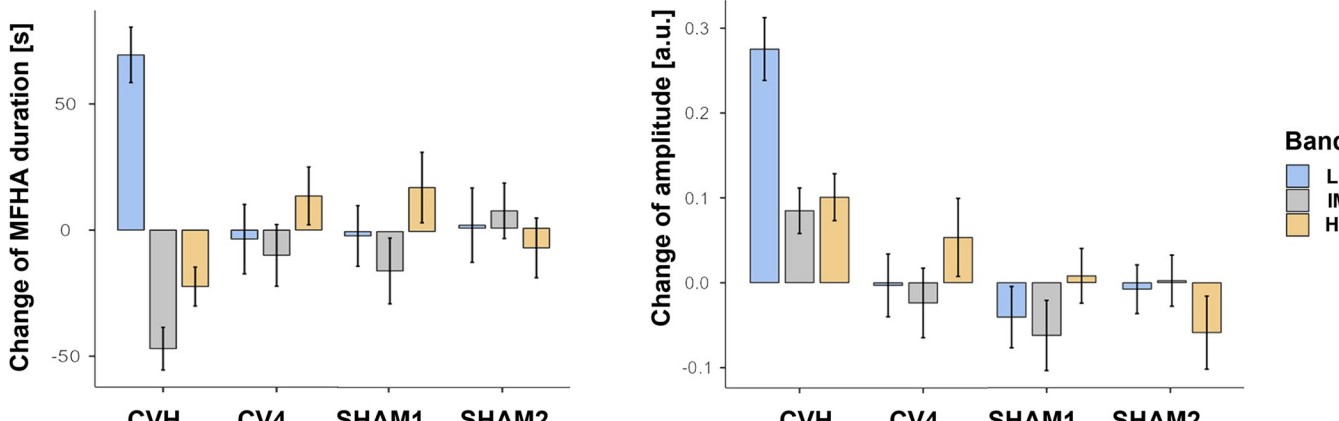

**Fig 4.** *Change of PPG MFHA durations (left) and photoplethysmography (PPG) amplitudes (right) from section 2 (eyes closed; EC1) to section 3 (Intervention).* The plots show most prominent changes only due to CVH for amplitudes and MFHA durations. There is a positive staircase (increase in both amplitude and MFHA [frequency] duration) for LF band, whereas there is a negative staircase for IM and HF band (amplitudes increase while MFHA [frequency] decrease). In CV4, there is a positive staircase for HF.

## PPG signal amplitudes

In a linear mixed-effects model, the *PPG amplitudes* were investigated with respect to the type of intervention, section, frequency band, and time (Fig 3B; Table 4 top). The main effects for band ($F(2, 1800.2) = 8.2$, $p < .001$) and section ($F(4, 1800.2) = 2.5$, $p < .05$) yielded significant effects, whereas the main effects for intervention ($F(3, 76.2) = 1.7$, $p = .17$), time ($F(1, 1856.1) = .3$, $p = .56$) and the covariate age ($F(1, 71) = 1.0$, $p = .32$) were not significant. There was further a significant interaction of intervention*section ($F(12, 1800.2) = 7.8$, $p < .001$), intervention*band ($F(6, 1800.2) = 6.1$, $p < .001$) and for intervention*section*band ($F(24, 1800.2) = 1.6$, $p = .03$), whereas the interaction band*section ($F(8, 1800.2) = 1.8$, $p = .08$) was not significant. Simple effects analyses showed a significant effect of section for CVH in the LF band ($F(4,1800) = 34.6$, $p < .001$), IM band ($F(4, 1800) = 3.2$, $p = .01$), and HF band ($F(4,1800) = 5.0$, $p < .001$), but not for CV4, SHAM1, and SHAM2 in either band (all $p > .1$). To investigate the specific effect of the section on participants who received the CVH intervention, Bonferroni-corrected post-hoc tests were computed for each frequency band. For the LF, IM, and HF bands, these tests showed a significant change in PPG amplitude for CVH from sections 1 to 3 (all $p < .001$), 2 to 3 (all $p < .001$), 3 to 4 (LF: $p < .001$; IM and HF: $p < .01$), 3 to 5 (LF: $p < .001$; IM: $p = .02$; HF: $p < .01$), but not from sections 2 to 4 and 1 to 5 (all $p > .5$).

To further illustrate this effect, *amplitude change* (Intervention–EC1) was computed for each intervention (Fig 4, right; Table 4, bottom). In a linear mixed-effects model, the change in amplitude was compared between interventions, frequency bands, and the intervention*band interaction with participants and examiners as random intercepts. There was a significant main effect of the intervention ($F(3, 72.8) = 6.2$, $p < .001$) and a significant intervention*band interaction ($F(6,301.8) = 4.5$, $p < .001$). The main effect of band was not significant ($F(2,301.8) = 2.3$, $p = .1$) as well as the covariate age ($F(1, 71.0) = 1.2$, $p = .27$). Simple effects were significant for the effect of intervention for the LF ($F(3, 164) = 12.7$, $p < .001$), but not for the IM ($F(3, 164) = 1.7$, $p = .17$), and HF bands ($F(3, 164) = 2.3$, $p = .08$). Bonferroni-corrected post-hoc tests further revealed a significantly higher amplitude change for CVH than for all other interventions in the LF band (all $p > .001$), but not for IM or HF (all $p > .5$).

## Electrocardiogram data

**Intervention effects on heart rate variability measures.** The effects of CVH, CV4, SHAM1, and SHAM2 were further investigated using a range of standard HRV measures derived from ECG data. HRV measures included RMSSD, the natural logarithm of power in different frequency bands (LF, IM, and HF), and the mean HR (Fig 5).

A linear effects model of the HR showed a significant effect of intervention ($F(3, 111.5) = 7.9$, $p < .001$), section ($F(4, 486.1) = 6.0$, $p < .001$) and intervention*time ($F(3, 493.0) = 14.0$, $p < .001$).

However, the covariate age ($F(1, 69.9) = 0.1$, $p = .72$), the main effect of time ($F(1, 491.9) = 0.7$, $p = .4$), the interaction of section*intervention ($F(12, 486.1) = 0.7$, $p = .8$), section*time ($F(4, 486.1) = 0.1$, $p = .98$) were not significant. These results indicate that there was a similar pattern of HR change across measurements, however, at different baseline levels. Bonferroni-corrected post-hoc tests did not show any significant changes across sections for any intervention.

A linear mixed effects model was used to examine RMSSD after logarithmic transformation (lnRMSSD). There was a significant effect of intervention ($F(3, 111.9) = 16.9$, $p < .001$), section ($F(4, 486.1) = 4.5$, $p < .01$), age ($F(1, 72.7) = 9.4$, $p < .01$), and a significant interaction of intervention*section ($F(12, 486.1) = 2.0$, $p < .05$) and intervention*time ($F(3, 493.3) =$, $p < .05$). Neither the effect of time was significant ($F(1, 492.4) = 1.2$, $p = .27$) nor the interaction of section*time ($F(4, 486.1) = 0.23$, $p = .9$) or intervention*section*time ($F(12, 486.1) = .6$, $p = .8$).

**Table 4. PPG amplitude analysis.** (Top) Mean PPG amplitude values for each intervention (CVH, CV4, SHAM1, SHAM2), frequency band (LF, IM, HF) and section (1 to 5). (Bottom) Mean change of PPG amplitude (±SD) for each intervention and frequency band.

| | | Section | Intervention | | | |
|---|---|---|---|---|---|---|
| | | | CVH | CV4 | SHAM1 | SHAM2 |
| Frequency band | LF | 1 | .49 (±.03) | .58 (±.04) | .57 (±.04) | .60 (±.04) |
| | | 2 | .43 (±.03) | .53 (±.04) | .49 (±.04) | .60 (±.04) |
| | | 3 | .68 (±.03) | .53 (±.04) | .47 (±.04) | .60 (±.04) |
| | | 4 | .45 (±.03) | .55 (±.04) | .53 (±.04) | .60 (±.04) |
| | | 5 | .43 (±.03) | .55 (±.04) | .55 (±.04) | .58 (±.04) |
| | IM | 1 | .56 (±.03) | .60 (±.04) | .59 (±.04) | .60 (±.04) |
| | | 2 | .54 (±.03) | .60 (±.04) | .57 (±.04) | .60 (±.04) |
| | | 3 | .62 (±.03) | .58 (±.04) | .54 (±.04) | .58 (±.04) |
| | | 4 | .55 (±.03) | .58 (±.04) | .55 (±.04) | .58 (±.04) |
| | | 5 | .58 (±.03) | .57 (±.04) | .59 (±.04) | .57 (±.04) |
| | HF | 1 | .52 (±.03) | .53 (±.04) | .52 (±.04) | .57 (±.04) |
| | | 2 | .52 (±.03) | .51 (±.04) | .50 (±.04) | .60 (±.04) |
| | | 3 | .62 (±.03) | .57 (±.04) | .53 (±.04) | .55 (±.04) |
| | | 4 | .56 (±.03) | .55 (±.04) | .55 (±.04) | .63 (±.04) |
| | | 5 | .56 (±.03) | .57 (±.04) | .59 (±.04) | .56 (±.04) |
| | | **Mean change of PPG amplitude** | **Frequency band** | | | |
| | | | | LF | IM | HF |
| | | Intervention | CVH | .28 (±.28) | .08 (±.20) | .10 (±.21) |
| | | | CV4 | .0 (±.20) | -.02 (±.22) | .05 (±.25) |
| | | | SHAM1 | -.04 (±.16) | -.06 (±.18) | .0 (±.14) |
| | | | SHAM2 | .0 (±.15) | .0 (±.16) | -.05 (±.23) |

Simple effects indicated that there was a difference between sections only for CV4 ($F(4, 486) = 7.15, p < .001$). Bonferroni-corrected post-hoc tests showed that for CV4, sections 1 and 4 ($p = .02$) as well as sections 1 and 5 ($p = .007$) showed significant increases. Age was also a significant factor for lnRMSSD, and post-hoc correlation analysis showed a significant negative correlation between lnRMSSD and age ($r = -.38, p < .001$). RMSSD was correlated with age, and the effect induced by our interventions was most prominent in the CV4 intervention, where the intervention was followed by an increase in lnRMSSD values in sections 4 and 5.

The power in different frequency bands was computed, and three separate linear mixed-effects models were used to investigate the effect of fixed and random factors on LF, IM, and HF power after logarithmic transformation (lnLF, lnIM, and lnHF, respectively). For lnLF power (0.05–0.12 Hz), there was a significant effect of section ($F(4, 486.3) = 3.7, p = .006$). All other effects did not reach significance: intervention ($F(3, 96.5) = 1.5, p = .22$), time ($F(1, 501.9) = 0.0, p = .99$), covariate age ($F(1, 43.4) = 3.3, p = .08$), the interactions intervention*section ($F(12, 486.3) = 1.7, p = .06$), intervention*time ($F(3, 503.4) = 2.5, p = .06$), section*time ($F(4, 486.3) = 0.3, p = .88$) and intervention*section*time ($F(12, 486.3) = 0.7, p = .75$). Bonferroni-corrected post-hoc tests for sections showed a significant difference between sections 1 and 4 ($p < .05$). Further, there was a negative relationship between lnLF power and age ($r = -.24, p < .001$).

In a linear mixed model of lnIM (0.12–0.18 Hz) power, there was a significant effect of section ($F(4, 486.3) = 2.8, p = .02$) and time ($F(1, 501.3) = 8.2, p < .01$), whereas the effects of intervention ($F(3, 97.5) = .9, p = .45$), age ($F(1, 63.0) = 2.7, p = .1$), the interaction of intervention*section ($F(12, 486.3) = 1.6, p = .08$), intervention*time ($F(3, 502.6) = 1.3, p = .28$), section*time ($F(4, 486.3) = 1.0, p = .38$) and intervention*section*time ($F(12, 486.3) = 1.1, p = .3$)

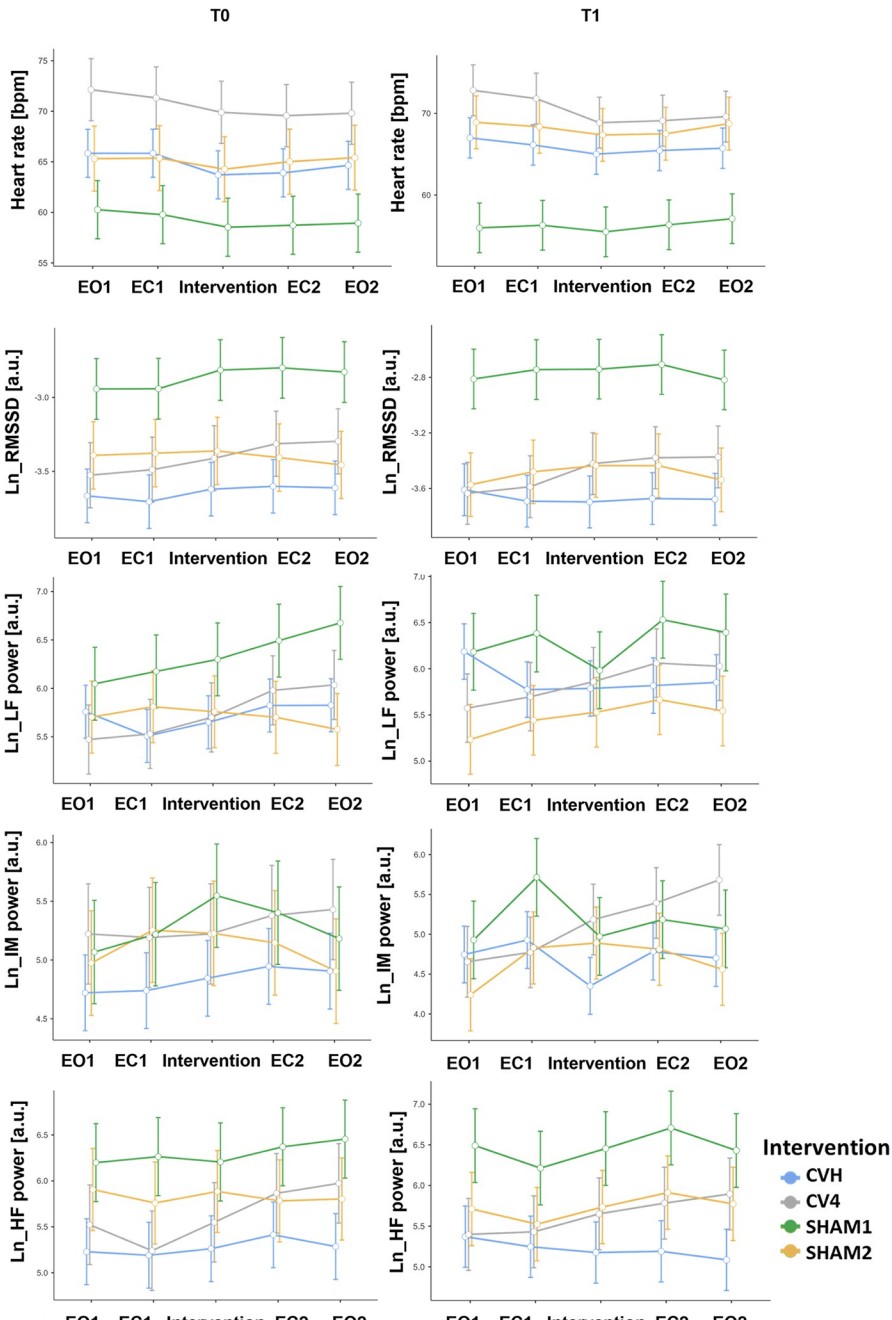

**Fig 5. Heart rate variability.** Effects of intervention, section and timepoint of measurement on electrocardiogram derived measures of heart rate variability (heart rate, RMSSD, LF/IM/HF power).

were not significant. Bonferroni-corrected post-hoc tests showed a significant increase from sections 1 to 4 ($p = .02$), but not for any other contrast.

Another linear mixed model for lnHF (0.18–0.4 Hz) power revealed significance for intervention ($F(3, 92.3) = 4.3$, $p = .008$), section ($F(4, 496.2) = 4.3$, $p = .002$), and age ($F(1, 67.8) = 14.1$, $p < .001$), but not for time ($F(1, 509.7) = .3$, $p = .6$) and intervention*section ($F(12, 496.1) = 1.2$, $p = .27$). Post-hoc tests showed a difference between CVH and SHAM2 ($p = .004$), but

not between any other interventions. Furthermore, there was only a significant difference between sections 2 and 4 ($p = .003$) and sections 2 and 5 ($p = .01$) for CV4.

## Discussion

We recently reported on successful manual in-phase palpation of physiological rhythms in changes in skin blood volume by experienced osteopathic examiners [21]. We found that these physiological rhythms are comparable to those reported in a comprehensive body of literature on PRM/CRI activity [27]. Importantly, our analysis of these data revealed compelling evidence for the inclusion of an intermediate (IM) frequency band, situated between the commonly used low-frequency (LF) and high-frequency (HF) bands, when examining autonomic involvement [9,12,28]. This IM band activity, originating in the reticular formation of the lower brainstem, is considered a peripheral manifestation of central network activity and is thought to play a crucial role in higher neural interoception.

### Current objective and results

In this study, we investigated the specificity of a standardized manually applied osteopathic cranial technique, the cranial vault hold (CVH), by comparing its effects to those of another established treatment, the compression of the fourth ventricle (CV4), and two sham interventions. The sham interventions were designed to mimic the physical aspects of CVH and CV4 without the intended therapeutic input. Our findings revealed significant CVH-specific autonomic nervous system (ANS) responses in skin blood volume (SBV) changes, in cardiac (heart rate, HR) measures, and delayed in respiration movements.

Quantified physiological responses in the momentary frequency of the highest amplitude (MFHA), in frequency band duration, and signal amplitude of participant's forehead PPG data before, during, and following CVH, CV4, SHAM1, and SHAM2 were investigated. The PPG data revealed a significant decrease in MFHA frequency during CVH but not in any other condition. This decrease in MFHA frequency with the highest amplitude in the LF range during CVH was accompanied by a significant increase in MFHA duration in the LF range and a significant decrease in MFHA duration in the IM band. Change scores (change in MFHA duration from section 2 (eyes closed) to section 3 (intervention)) showed a larger change in CVH compared to all other interventions in the LF band, and a smaller change in the IM band for CVH compared to SHAM2. Our results further showed that decreased frequencies during CVH returned to baseline during the following resting conditions (eyes closed and eyes open without intervention). Therefore, the observed changes in PPG signals seem to be specific to the CVH intervention.

The significant effects seen in MFHA (durations) could also be found PPG signal amplitudes. Our data showed a significant concomitant increase of LF, IM and HF amplitudes during CVH, but not in any other intervention. Respiratory frequencies computed from the MFHA exhibited no significant change across sections in any group, therefore suggesting that effects seen in PPG signals were not induced by specific respiratory changes. However, data published in our previous publication suggested a synchronization of 1:2 or 1:3, respectively, between PPG and respiration frequencies during CVH [16]. In HRV data there was a significant effect of intervention for heart rate indicating different baseline values without specific changes in any intervention as well as a trend of decreasing heart rate across sections. After logarithmic transformation, lnRMSSD showed a significant increase across sections. Simple effects showed that this effect was specifically found in the CV4 condition and indicates an increase of parasympathetic tone. As expected, age was also a significant factor for lnRMSSD, and correlation analyses showed a significant negative correlation between lnRMSSD and age

($r$ = -.38, $p < .001$), which is in line with previous findings [29]. The power after logarithmic transformation in the LF, IM, and HF bands did not seem to be specific to any intervention. The ECG data showed that the strong effects in the CVH group seen in PPG signals could not be replicated in HRV measures (heart rate, RMSSD, power in LF, IM and HF bands) retrieved from ECG data.

Our current findings indicate that the CVH elicits a specific systemic physiological response mediated through the ANS, rather than merely a localized reaction. This conclusion is supported by the observed changes in ANS activity, primarily reflected in the MFHA measurements. The specificity of the CVH intervention is particularly evident in the amplitudes of skin blood volume (SBV) changes. These alterations can be attributed to the unique effects of CVH on ANS activity, distinguishing it from other interventions studied. However, it is important to note that the detection of these effects was complicated by substantial individual variability. This high variance may explain why adaptations to interventions within other physiological systems were not readily apparent in frequency or amplitude parameters. Preliminary analyses using coordination algorithms (data not shown) lend further support to this interpretation.

In summary, our results provide compelling evidence that physiological responses to CVH are systemically mediated through changes in ANS activity. These changes are most prominently observed in MFHA measurements and SBV amplitude alterations. This suggests that CVH has a distinct impact on autonomic regulation compared to other osteopathic techniques and sham interventions.

## Physiological backgrounds

Recent physiological evidence has revealed significant changes in ANS activity within the LF- and IM-bands during CVH stimulation [21]. These changes align with the frequency ranges associated with the primary respiratory mechanism (PRM) or cranial rhythmic impulse (CRI), respectively. Furthermore, the rhythmic response patterns observed in SBV changes and respiration during CVH correspond to previously identified reticular '0.15 Hz rhythm band' activity. In our previous publication, a cluster analysis displayed two general systemic response patterns: one dominated by IM activity (47% of participants) responding to CVH with a highly stable oscillation at a mean at 0.08 Hz, and one with dominant LF activity exhibiting increases to IM activity following CVH. Notably, stable respiration-PPG frequency ratios of approximately 3:1 at the second day of measurement suggest that respiration was not the primary driver of enhanced oscillations in PPG during CVH. However, the potential involvement of baroreflex pathways in generating these SBV oscillations during CVH cannot be ruled out. This hypothesis is supported by the observation that frequencies around 0.08 Hz displayed an almost tonic course.

The interpretation of cardiovascular-respiratory time series data is complicated by the presence of certain frequencies that may represent subharmonics of primary IM band activity, such as 0.16 Hz and 0.08 Hz. This presents a challenge in data analysis, as the lower frequencies observed in our current study could potentially originate from baroreceptor activity rather than being true subharmonics. Our previous research [30] on autogenic training (AT) provides additional context for understanding IM band activity. We found distinct adaptation patterns in the IM band among healthy subjects engaged in AT. Notably, AT-naïve participants exhibited spontaneous IM activity at approximately 10%, while AT experts demonstrated this activity at an average of 70%. This complexity necessitates careful consideration when drawing conclusions from observed frequency patterns in cardiovascular-respiratory time series.

Our current study incorporates and expands upon CVH data presented in our earlier publication, providing a more comprehensive analysis of physiological responses. We demonstrate that the observed reactions in PPG data are highly specific to CVH, as evidenced by their absence during CV4 or any comparable SHAM condition. This finding significantly strengthens the validity and specificity of our results. Previous studies in the field of HRV have often been limited by large variances in data, as noted by [31]. To address this challenge, we introduced a methodological innovation in our PPG data analyses. Specifically, we included an Intermediate (IM) frequency band alongside the commonly applied LF and HF bands. This approach has proven instrumental in enhancing our ability to identify and characterize specific physiological reactions during CVH.

Reticular activity around 0.15 Hz was initially observed in animal models by Langhorst and Lambertz. Recent fMRI studies have confirmed similar pacemaker activity in the left human brainstem [32], validating these earlier findings. Langhorst and Lambertz's research on 400 neurons demonstrated that constant rhythmic activity in this frequency band first emerged in unspecific reticular neurons during decreased activation or arousal. This process subsequently led to "entrainment" of the entire cardiovascular-respiratory system at this rhythm [8]. These findings collectively highlight the importance of reticular activity in regulating cardiovascular and respiratory rhythms not only in animal models but also in humans.

Of note, this rhythm is not identical with the rhythm of respiration, although both rhythms were observed to coincide. Perlitz and colleagues [30] noted that within the typical reticular rhythm (retR) frequency range (as observed in animal experiments), relaxed subjects showed amplified modulations in heart rate, blood pressure, and electrodermal activity, even with controlled respiration frequencies [33,34]. These dynamics are fundamental for the activity in the 0.15 Hz rhythm band, which is particularly evident in different ratios of synchronization between respiration and other peripheral physiological systems.

This, however, is of paramount importance for osteopathy since experienced investigators in the field can manually palpate the PRM/CRI throughout the body. Thus, hitherto held explanations of rhythmic pulsations of bony and connective tissue structures need not contradict IM band physiology and its reticular rhythms, which may also entrain connective tissues. This is suggested by recent reports on dynamic phasic responsiveness of contractile properties of connective tissues [35]. Furthermore, anatomical findings cited 20 years ago demonstrated that numerous bony structures were not ossifying during life but remained elastic and flexible along with connective tissues, vessels, nerves, and sutures [1]. In light of these findings, we hold that IM activity may not only entrain connective tissue structures but that they may augment the reticular rhythm. When emerging in cerebrospinal fluid (CSF), this rhythm may set off oscillations of these frequencies thereby establishing closed-loop feedback with those areas in the lower brainstem responsible for generating IM activity. This concept aligns with Sutherland's proposition that CSF waves transfer movements from the cranium to the sacrum, although he was unaware of the reticular formation's rhythm-generating neural activity. CVH, as a standard osteopathic hands-on technique, appears effective in seizing and triggering these processes relevant for the maintenance and augmentation of these rhythms in the lateral cranial regions and cranial bones. Physiological baroreflex activity generated in the dorsal medial nucleus tractus solitarii (dmNTS) reflex arcs is anatomically adjacent to reticular formation structures, just as they are functionally connected by adjacent or overlapping rhythmic activity. Such relatedness of oscillations may (further) augment them [36].

Recent attempts to establish a comprehensive understanding of the relationships between various tissues have fallen short in fully incorporating the diverse neural sources of rhythmic activity. For instance, [37] referred to the Intermediate (IM) band activity solely as a function of the sympathetic nervous system. Such interpretations fall short of comprehensively

understanding the plethora of experimental findings on the physiology and neurobiology underlying these rhythms.

Our findings offer new insights into the intricate interplay between CVH, ANS activity, and physiological rhythms. These discoveries underscore the need for further investigation into the underlying mechanisms and potential clinical applications of CVH. Given the critical role of ANS involvement in numerous serious medical conditions, CVH's ability to trigger profound ANS responses warrants special attention. We propose that this technique should be given greater emphasis in medical curricula, particularly considering its potential as a non-invasive method for modulating ANS activity.

## Limitations

This study expanded our previous research on CVH physiology by employing a between-subjects design, including CV4 and two comparable SHAM interventions. However, several limitations warrant careful consideration when interpreting our findings. Physiological signals inherently exhibit significant variability due to multiple interconnected subsystems. Averaging across measurements and participants may not only fail to account for this variance but could potentially suggest false activities. Our sample comprised groups of unequal sizes, though we carefully applied appropriate statistical analyses to mitigate this shortcoming. Further, we observed significantly different baseline heart rates between groups, which could be a confounding factor. To address this, we employed linear mixed effects analyses using participant and examiner as random intercepts. Additionally, we conducted analyses using the first section as a baseline covariate, yielding largely similar results to those reported. To fully comprehend the specificity of CVH findings, future studies should incorporate analyses of dynamic interactions within participants and between examiners and participants. While adding the IM frequency band offers a better fit within the widely disputed LF range [14], it does not entirely resolve the issue of rigid frequency bands. Lastly, our analyses rely on MFHA and TFD methods, which have inherent limitations. These include the suppression of minor amplitude frequencies and some frequency uncertainty, particularly at lower frequencies. These limitations highlight areas for improvement in future research and should be considered when interpreting the study's results.

## Conclusions

This study demonstrates that the cranial osteopathic intervention known as Compression of the Fourth Ventricle (CVH) significantly triggers specific Autonomic Nervous System (ANS) responses in both the primary and secondary Intermediate (IM) frequency bands of Skin Blood Volume (SBV). These responses likely originate from either enhanced reticular activity or the nucleus tractus solitarii (NTS) reflex arcs and associated baroreflex activity. The involvement of these crucial neural structures provides a plausible physiological basis for the broad-spectrum beneficial effects often attributed to Osteopathy in the Cranial Field (OCF). Our findings offer valuable insights into the neurophysiological mechanisms underlying CVH and its effects on autonomic function. The observed changes in ANS activity, particularly in the IM frequency bands, suggest that CVH may have far-reaching implications for various physiological processes regulated by the autonomic nervous system. The implication of central nervous system structures known to play vital roles in autonomic regulation strengthens the scientific rationale for the use of CVH and other OCF techniques in clinical practice. These results not only validate the physiological impact of CVH but also provide a foundation for understanding its diverse therapeutic benefits. Future research should focus on elucidating the precise mechanisms by which CVH influences ANS activity, exploring the clinical implications of these

findings across various patient populations, and investigating the potential long-term effects of repeated CVH interventions on autonomic function. In conclusion, this study marks a significant step forward in understanding the neurophysiological basis of OCF techniques and their potential therapeutic applications. By demonstrating the specific ANS responses triggered by CVH and linking them to key neural structures, we provide a solid foundation for further research and clinical application of this osteopathic intervention.

## Author Contributions

**Conceptualization:** Volker Perlitz, Holger Pelz, Klaus Mathiak.

**Data curation:** Micha Keller, Holger Pelz, Gero Müller.

**Formal analysis:** Micha Keller, Volker Perlitz, Stefan Borik, Birol Cotuk, Gero Müller.

**Investigation:** Holger Pelz, Ines Repik, Armin Geilgens, Johannes Mayer.

**Methodology:** Micha Keller, Volker Perlitz, Gero Müller, Klaus Mathiak.

**Project administration:** Holger Pelz.

**Software:** Stefan Borik, Gero Müller.

**Supervision:** Volker Perlitz, Holger Pelz.

**Writing – original draft:** Micha Keller, Volker Perlitz.

**Writing – review & editing:** Micha Keller, Volker Perlitz, Holger Pelz, Stefan Borik, Birol Cotuk, Gero Müller, Klaus Mathiak.

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
