## [Decision Letter · Decision Letter 0]

1 Dec 2024

PONE-D-24-41518Specificity of Cranial Cutaneous Manipulations in Modulating Autonomic Nervous System Responses and Physiological Oscillations: A Controlled StudyPLOS ONE

Dear Dr. Keller,

Thank you for submitting your manuscript to PLOS ONE. After careful consideration, we feel that it has merit but does not fully meet PLOS ONE’s publication criteria as it currently stands. Therefore, we invite you to submit a revised version of the manuscript that addresses the points raised during the review process.

We look forward to receiving your revised manuscript.

Kind regards,

Sanaullah Sajid, M.Phil/PhD

Academic Editor

PLOS ONE

Journal Requirements:

2. Thank you for stating the following financial disclosure: [HEAD Genuit Stiftung (Aachen, Herzogenrath), Grant/Award Number: HGS03S18072016].

3. In the online submission form, you indicated that [The data that support the findings of this study are available from the corresponding author, MK, upon reasonable request].

Reviewers' comments:

Reviewer's Responses to Questions

**Comments to the Author**

1. Is the manuscript technically sound, and do the data support the conclusions?

Reviewer #1: Yes

2. Has the statistical analysis been performed appropriately and rigorously? 

Reviewer #1: Yes

3. Have the authors made all data underlying the findings in their manuscript fully available?

Reviewer #1: No

4. Is the manuscript presented in an intelligible fashion and written in standard English?

Reviewer #1: Yes

5. Review Comments to the Author

Reviewer #1: Firstly, I would like to express my gratitude for your beautiful and innovative research. My comments are as follows:

1-despite all examiners having sufficient experience, doesn't the significant difference in the processes conducted by Examiner 1 compared to other examiners cast doubt on the accuracy of the obtained data?

2- There are 13 articles cited as references in the manuscript that date back to before 2005. It would be better, if possible, to replace them with more recent references.

3-The keywords used should be reviewed for compliance with MeSH (Medical Subject Headings) standards, and if possible, fewer keywords should be used.

4-Considering that a large amount of data has been presented in the article, and one of the important aspects of publishing an article is ensuring it is easily understandable for readers, it is recommended, at the discretion of the esteemed editor, to include a graphical abstract that presents the process and overall results of the work in the article

5-If possible, please use a different chart format to present the data in Table 3. The data dispersion has led to significant fluctuations in the table, and the current format does not effectively convey the results.

6. PLOS authors have the option to publish the peer review history of their article (what does this mean?). If published, this will include your full peer review and any attached files.

Reviewer #1: No

---

## [Author Response · Author response to Decision Letter 0]

19 Dec 2024

Please find all updated figures in the attached files

The reference style has been changed to Vancouver style (only in final manuscript). Furthermore, a sentence has been added concerning data availability. The data will also be accessible through the DGOM, e.V. which has indirectly been involved in the study and will store the orignal data.

---

## [Editor Report · Decision Letter 1]

26 Dec 2024

Specificity of Cranial Cutaneous Manipulations in Modulating Autonomic Nervous System Responses and Physiological Oscillations: A Controlled Study

PONE-D-24-41518R1

Dear Dr. Keller,

We’re pleased to inform you that your manuscript has been judged scientifically suitable for publication and will be formally accepted for publication once it meets all outstanding technical requirements.

Kind regards,

Sanaullah Sajid, M.Phil/PhD

Academic Editor

PLOS ONE
---

## [Editor Report · Acceptance letter]

8 Jan 2025

PONE-D-24-41518R1 

PLOS ONE

Dear Dr. Keller, 

I'm pleased to inform you that your manuscript has been deemed suitable for publication in PLOS ONE. Congratulations! Your manuscript is now being handed over to our production team.

Kind regards, 

on behalf of

Dr. Sanaullah Sajid 

Academic Editor

PLOS ONE